## RESEARCH ARTICLE

# How octopuses use and recruit additional arms to find and manipulate visually hidden items

Ashley J. Gendreau[1,*], Camille Boucaud[1,2], Kendra C. Buresch[1], Allison S. Rooney[1,3], Halia Morris[1,4] and Roger T. Hanlon[1]

## ABSTRACT

Shallow-water benthic octopuses are tactile creatures that explore their environment mainly using chemotactile receptors in their suckers. Details of how they coordinate multiple arms to sense and manipulate items remain unknown. We developed a behavioral assay that exploits their natural foraging behavior to quantify how octopuses use and recruit their arms when searching for and investigating items in a visually occluded environment. Fourteen *Octopus bimaculoides* were presented with an opaque rock dome to 'blindly' explore for one of five items placed inside. During 117 experimental trials, 2327 arm actions and 394 coordinated arm recruitments were video recorded. Octopuses most often recruited the nearest-neighboring arm when manipulating items (44%). Recruitment of more distant arms was collectively observed (56%). The most common recruitment patterns were (1) initial arm→nearest neighbor→nearest neighbor and (2) initial arm→nearest neighbor→second arm over. We discovered that octopuses used all eight arms with similar frequency and most often engaged three, four, or five arms simultaneously. These findings further demonstrate the flexibility and functionality of all eight arms and indicate that octopuses can use all eight arms equally, which may inform research into arm neuroanatomy as well as the design of coordinated soft robotic arms.

KEY WORDS: Octopus, Arm recruitment, Behavioral flexibility, Behavioral ecology, Sensory ecology, Bioinspiration

## INTRODUCTION

Octopuses are mainly tactile feeders, but they integrate multiple senses including vision, taste, and touch to perform a variety of tasks. While octopuses indeed have keenly developed vision (Young, 1971; Wentworth and Muntz, 1992; Hanke and Kelber, 2020), most of their food is hidden amidst coral, algae, seagrass or rock rubble. They use their acute vision to move to likely food spots then 'blindly' insert various arms into crevices to detect, capture, and retrieve live food items such as shrimp, clams, crabs, and fishes (Ambrose and Nelson,

[1]Bell Center, Marine Biological Laboratory, Woods Hole, MA 02543, USA. [2]Michigan State University, Department of Microbiology, Genetics, and Immunology, East Lansing, MI 48824, USA. [3]Northeastern University, College of Science, Boston, MA 02115, USA. [4]Hampton University, Department of Marine and Environmental Science, Hampton, VA 23668, USA.

*Author for correspondence (agendreau@mbl.edu)

A.J.G., 0009-0002-2304-149X; C.B., 0000-0003-3437-7028; K.C.B., 0000-0002-0969-4008; A.S.R., 0009-0006-0715-1997; H.M., 0009-0000-5961-1736; R.T.H., 0000-0003-0004-5674

1983; Forsythe and Hanlon, 1997; Leite et al., 2009; Mather et al., 2014; Hanlon and Messenger, 2018; Maselli et al., 2020). Each of their eight hyper-redundant and flexible arms (Mather, 1998; Kier, 2016; Kennedy et al., 2020) is lined with hundreds of suckers, which each contain thousands of sensory cells (Graziadei, 1971) that serve as the primary means for acquiring mechanosensory (Wells, 1962) and chemosensory information (Graziadei, 1964; Wells, 1962; Wells et al., 1965; Fouke and Rhodes, 2020). These senses guide live prey capture and handling, allowing octopuses to successfully identify, immobilize and eat live prey without visual stimuli (Kier and Smith, 1990; Hanlon and Messenger, 2018; Fouke and Rhodes, 2020; Buresch et al., 2024).

Control of this sensory information is upregulated by a partially decentralized nervous system that allows for some local peripheral coordination amongst arms as well as overall sensorimotor coordination by the central brain (Young, 1971; Hochner, 2012). Approximately 330 million of the 500 million neurons in an octopuses' nervous system (Young, 1963) are distributed throughout the peripheral nervous system (PNS), which can independently coordinate some inter-arm sensing and actions (Altman, 1971; Sumbre et al., 2001; Gutnick et al., 2011; Kuuspalu et al., 2022; Chang and Hale, 2023).

While each arm has equal potential of accomplishing the same tasks, some evidence points to limb specialization and partitioning of arms for particular tasks (Mather, 1998; Byrne et al., 2006a). Research indicates preferential and repeatable arm recruitment sequences between coordinated arms, such as the employment of adjacent arms during visual prey attacks and item manipulation (Byrne et al., 2006b; Bidel et al., 2022). Octopuses also occasionally exhibit labor division between the anterior and posterior arms during specific tasks, using posterior arms for walking (Mather, 1998) and anterior arms when reaching for prey (Byrne et al., 2006a,b) and in visually guided prey attack (Bidel et al., 2022). However, the ways by which octopuses coordinate multiple arms or deploy individual arms for exploring in visually occluded settings – as they do in natural environments – remain largely uncharacterized.

We developed a live-animal behavioral assay that exploited the natural arm searching behavior of *Octopus bimaculoides*. Our approach allowed us to characterize patterns of individual arm use and recruitment between coordinated arms as octopuses explored and manipulated a variety of prey and non-prey items in a visually occluded environment. Our objectives were to (1) describe general arm behavior during exploration and contact with items, (2) assess if individual arms were used with similar frequency, (3) determine the most common type of arm recruitment, (4) identify common patterns of arm recruitment during the coordination of multiple arms, and (5) quantify the number of arms used simultaneously to explore and manipulate items.

## RESULTS

A total of 2327 individual arm actions (which included 1607 incidences of arm exploration and 720 incidences of contact with

the item inside of the dome) and 394 incidences of arm recruitment were recorded from a total of 117 experimental trials. Octopuses approached the opaque dome (*n*=630 total approaches) from various angles (e.g. from above, left, or right) and searched inside of the dome multiple times per trial (mean number of approaches per trial=5). Octopuses used their arms to explore the dome's interior or contact the item inside or both, often using multiple arms in a seemingly haphazard sequence. When coordinating multiple arms inside of the dome, octopuses often employed three, four, or five arms simultaneously, while maintaining grip and stability on the dome's exterior with the remaining arms.

### Anterior arms were used for initial interactions

In the first approach to the dome in each trial, octopuses preferentially used one of their anterior arms, rather than one of their posterior arms, as the initial arm inserted into the dome (Fig. 1; Table S1a; anterior=88, posterior=29; Wilcoxon signed-rank test: $W$=76.0, $n$=13, $P$=0.006, exact two-tailed). However, there was no evidence of a preference for using a specific anterior arm (Fig. 1; Table S1a; R1=25, R2=22, L1=23, L2=18; Friedman's ANOVA: $\chi^2$=2.70, *d.f.*=3, $P$=0.441). In subsequent approaches within the same trial, anterior and posterior arms were used with equal frequency as the initial arm inserted into the dome (Table S1b; anterior=260, posterior=253; Wilcoxon signed-rank test: $W$=−1.0, $n$=14, $P$=0.989, exact two-tailed).

### Arm use during exploration

A total of 1607 incidences of arm explorations were recorded. During exploration of the dome's interior, all eight arms were used at similar frequency (Fig. 2A; Table S2; Friedman's ANOVA:

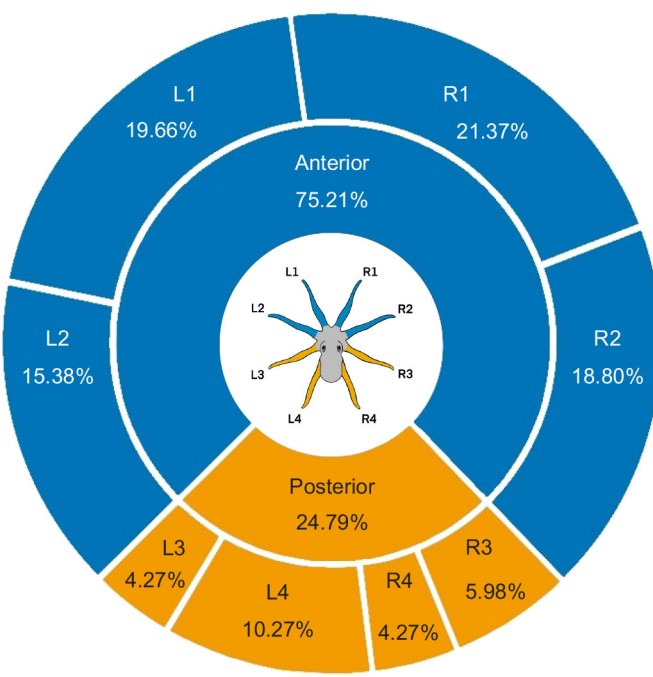

**Fig. 1. Initial arm inserted into the dome.** At the beginning of each experimental trial (*n*=117), octopuses (*n*=14 individual animals) most frequently used one of their four anterior arms as the initial arm inserted into the dome 75% of the time (blue rings: anterior=88 trials; orange rings: posterior=29 trials). Each insertion was recorded from a unique trial, and animal identity was tracked throughout. A significant anterior preference was detected (Wilcoxon signed-rank test, two-tailed: $W$=76.0, $P$=0.006). No significant difference was found among individual anterior arms (Friedman's ANOVA: $\chi^2$=2.70, *d.f.*=3, $P$=0.441).

$\chi^2$=10.47, *d.f.*=7, $P$=0.164). When exploratory arm use was grouped by body region, a modest anterior bias was detected (Wilcoxon signed-rank test: $W$=65.0, $n$=14, $P$=0.040, exact two-tailed), but this effect was eliminated when the initial arm used in the first approach of each trial was excluded (Wilcoxon signed-rank test: $W$=42.0, $n$=14, $P$=0.197), suggesting that the initial insertion bias was responsible for the apparent anterior preference. No significant difference was found between left and right exploratory arms (Wilcoxon signed-rank test: $W$=34.5, $n$=14, $P$=0.268, exact two-tailed).

Octopuses explored the dome's interior with a sweeping motion in 94.29% of approaches (*n*=594 approaches; *n*=1607 incidences of arm exploration). Exploration occurred before, during, and after contact with the item inside of the dome. The sweeping motion of the arm was only moderately effective in establishing immediate initial contact with the item (in 33.33% of all trials the initial arm inserted into the dome made immediate contact); in the remaining trials initial contact often occurred after one to three incidences of exploration (Table S3a; one exploration: 33.33%, two explorations: 28.21%, three explorations: 19.23%). After the first contact with the item, octopuses continued to explore the dome's interior with some arms, while other arms were used to manipulate the item inside of the dome. After contacting and manipulating the item, octopuses either dropped the item or, if the item was small enough to be removed, extracted it from the dome. Octopuses extracted the item from the dome in 89.74% of experimental trials with a removable item (*n*=78 trials). After an item was extracted from the dome, octopuses often returned to explore the dome's interior (Table S3b; average number of additional approaches=3.186). Following item extraction, octopuses averaged eight additional incidences of explorations per approach and frequently utilized two (37.50%) or three (28.57%) different arms to explore per approach (Table S3b).

### Arm use during contacting

A total of 720 incidences of arm contacts were recorded. During item contact and manipulation, arm use was not evenly distributed (Fig. 2B; Table S4; Friedman's ANOVA: $\chi^2$=25.25, *d.f.*=7, $P$<0.001). Post-hoc comparisons revealed that this effect was driven primarily by the consistent underuse of arm R4 and, to a lesser extent, arm R3 (see Table S5 for full pairwise statistics). The underuse of these two arms resulted in a significant bias toward anterior arms (Wilcoxon signed-rank test: $W$=71.0, $n$=14, $P$=0.023, exact two-tailed) and toward left arms (Wilcoxon signed-ranked test: $W$=−66.0, $n$=14, $P$=0.037, exact two-tailed).

### Arm recruitment during item contacting

When an item remained inside the dome, octopuses touched the item in 69.23% of all approaches (*n*=280 approaches). In these approaches, octopuses often recruited one or more additional arms to interact with the item, resulting in a total of 394 recruitment events (55.07% of approaches). In many approaches, octopuses contacted the item with only a single arm and did not recruit additional arms (43.93% of approaches). The nearest-neighboring arm (A+1) was the most frequently used recruitment type between any two arms used sequentially to contact the item (44%) (Fig. 3; Table S6). A Friedman's ANOVA revealed significant differences in the frequency of recruitment types ($\chi^2$=28.84, *d.f.*=3, $P$<0.001). Post-hoc Wilcoxon signed-rank tests showed that A+1 recruitments occurred significantly more often than all other recruitment types (collectively 56%) (all $q$<0.010, all $P$<0.010). Recruitments of the second arm over (A+2) occurred significantly more frequently than recruitments of the furthest arm away (A+4) ($q$=0.006, $P$=0.008), but not more often than the third arm over (A+3) ($q$=0.060, $P$=0.143). No significant difference was

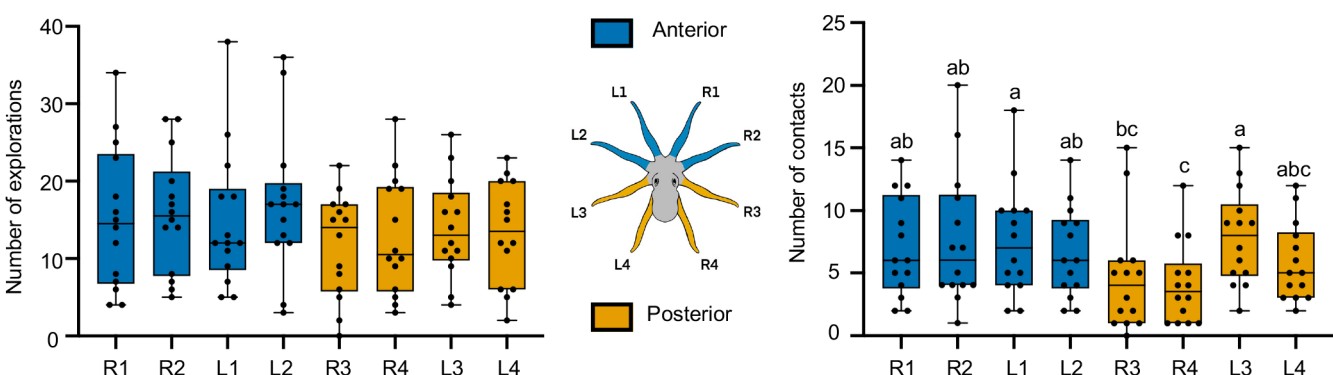

**Fig. 2. Use of individual arms for exploration and item contact.** (A) During exploration of the dome's interior, octopuses ($n$=14 individual animals; biological replicates) used all eight arms with similar frequency (Friedman's ANOVA: $\chi^2$=10.47, $d.f.$=7, $P$=0.164). No anterior-posterior bias ($W$=42.0, $P$=0.197) or left-right bias was observed ($W$=34.5, $P$=0.268). (B) In contrast, when contacting and manipulating the item inside of the dome, arm use differed significantly (Friedman's ANOVA: $\chi^2$=25.25, $d.f.$=7, $P$<0.001). This effect was driven by the underuse of arms R3 and R4, as shown by the compact letter display (CLD) groupings (letters 'a', 'b', and 'c' indicate statistically distinct groups; see Table S5 for full pairwise statistics). This resulted in a significant anterior bias ($W$=71.0, $P$=0.023) and left-arm bias ($W$=−66.0, $P$=0.037). All tests were two-tailed. Data are presented as box-and-whisker plots: horizontal lines indicate medians, box edges represent 25th and 75th percentiles, whiskers span minima to maxima, and black points represent total observations for each individual octopus.

observed between A+3 and A+4 recruitment types ($q$=0.085, $P$=0.242).

### Multiple arm recruitment patterns

Octopuses sequentially used three or more arms to interact with the item inside of the dome in 29.18% of approaches in which contact occurred ($n$=83 recruitment sequences of three or more arms) (Fig. 4; Table S7). Of the 16 possible recruitment patterns among the coordinated use of three arms, two patterns occurred significantly more frequently than expected by chance: initial contacting arm→nearest neighbor→nearest neighbor (A to A+1 to A+1) ($n$=22, 26.51%), and initial contacting arm→nearest-neighbor→second arm over (A to A+1 to A+2) ($n$=11, 13.25%) (Chi-square: $\chi^2$=81.71, $d.f.$=15, $P$<0.001). The remaining 14 patterns were each observed at least once (Fig. 4; Table S7).

### Simultaneous arm use

Octopuses most frequently used three to five arms simultaneously to explore the dome's interior or interact with the item inside, or both (Fig. 5; Table S8; three arms: $n$=32, 27.35%, four arms $n$=36, 30.77%, five arms $n$=36, 30.77%; Chi-square: $\chi^2$=134.6, $d.f.$=7, $P$<0.001). Post-hoc Wilcoxon signed-rank tests indicated that the frequencies of three-, four-, or five-arm use did not differ significantly from one another (all $q$=1.0, all $P$>0.566). Octopuses never used all eight arms inside of the dome simultaneously. Only one trial involved the simultaneous use of seven arms, and one trial involved the use of a single arm throughout. The simultaneous use of two or six arms was infrequent (Fig. 5; Table S8; two arms: $n$=6, 5.13%, six arms: $n$=5, 4.27%).

### DISCUSSION

When foraging for food, which sensory modality primarily guides the use and coordination of an octopus's eight arms: vision or chemotactile perception? This question is fundamental in the context of sensory ecology, and its answer provides useful insights into the strategies octopuses employ to locate prey during the brief periods that they leave their protective dens. While octopuses rely on vision to detect and capture certain prey, such as crabs in open environments, these instances are relatively uncommon in nature. Instead, octopuses use

vision primarily to identify potential foraging sites that may conceal live prey (e.g. mussels, crabs, shrimps, small fishes), and subsequent searching behavior often involves blind exploration within such sites (e.g. coral structures and rock crevices) (Forsythe and Hanlon, 1997; Leite et al., 2009; Mather et al., 2014; Hanlon and Messenger, 2018; Maselli et al., 2020).

In the laboratory experiments reported here, octopuses demonstrated unique behavioral flexibility, using and coordinating multiple arms to search and interact with items in a visually occluded environment. While all eight arms were engaged during exploration and contact, usage was not uniformly distributed. When recruiting additional arms during item manipulation, octopuses most frequently used nearest-neighboring arms in sequential contact, although more distant arms were often involved. Interestingly, octopuses tended to use three to five arms at a time to explore and manipulate items, rather than deploying all eight arms simultaneously. These findings are discussed in relation to the neuroanatomy of octopuses, ecologically significant behaviors, and the principles of biologically inspired soft robotics.

### No arm specialization during visually occluded foraging

Similarities in octopus arm anatomy (Graziadei, 1965, 1971; Kier and Stella, 2007; Kier, 2016) suggest that all eight arms are capable of functioning equivalently, with little specialization. That is, octopuses could use any of their eight arms, singularly or in combination, for locomotion, grooming, and manipulation of items or prey (Mather, 1998; Levy and Hochner, 2017). We observed this behavioral flexibility in our experiments: during both exploration and item manipulation, octopuses engaged and coordinated multiple arms with remarkable similarity. We did, however, detect a subtle but consistent underuse of the right posterior arms – R3 and R4 – during contacting behaviors. One possible explanation for this observation is that arm R3 is the hectocotylized arm used by male octopuses for mating, which could influence its behavioral role (Weertman and Scheel, 2024). Additionally, octopuses arms have been shown to act in tightly coordinated neighboring pairs (Chang and Hale, 2023), and the underuse of arm R3 may influence the engagement of its adjacent partner, R4. This interdependence between neighboring arms suggests that even subtle biases in the recruitment of a single arm could impact the involvement of nearby limbs during item interaction.

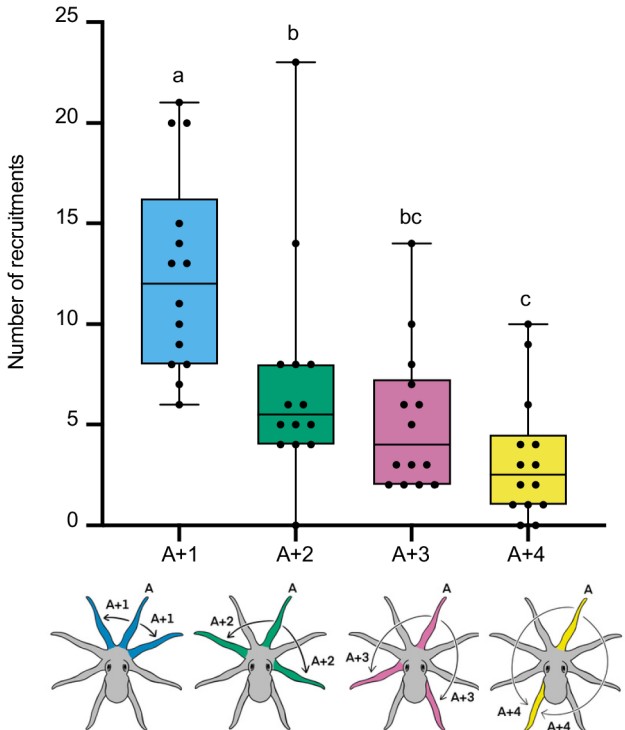

**Fig. 3. Additional arms were recruited after an arm initially contacted an item of interest.** When contacting the item inside of the dome, octopuses (*n*=14 individual animals; biological replicates) recruited additional arms (*n*=394 total recruitments) to assist in manipulation. The nearest-neighboring arm (A+1) was recruited significantly more frequently (44%) than the second arm over (A+2), third arm over (A+3) or farthest arm away (A+4) (collectively 56%) (Friedman's ANOVA: $\chi^2$=28.84, *d.f.*=3, *P*<0.001). Post-hoc Wilcoxon signed-rank tests [Benjamini, Krieger and Yekutieli (BKY) correction] revealed that A+1 recruitments occurred significantly more frequently than all other types (all adjusted *q*<0.010, all *P*<0.010). Data are presented as box-and-whisker plots: horizontal lines indicate medians, box edges represent 25th and 75th percentiles, whiskers span minima to maxima, black points represent total observations for each octopus. CLD groupings (letters 'a', 'b', and 'c') indicate statistically distinct groups. All tests were two-tailed.

While an anterior bias was evident in the first arm inserted into the dome during initial approaches, this bias did not persist throughout the trial, and no consistent left versus right lateralization emerged. This lack of specialized arm use appears to be unique to visually occluded environmental exploration and item manipulation. The general redundancy in arm use during visually occluded tasks likely confers several advantages, such as enhanced flexibility in movement and manipulation, enabling octopuses to navigate diverse environments and perform a wide range of tasks even when searching for prey in cluttered and visually challenging environments. Moreover, non-specialized arm usage may provide increased resilience; in the event of injury or damage to one arm, an octopus can continue to forage effectively using its remaining arms.

### Arm recruitment during visually occluded foraging
Octopuses in our experiments nearly always recruited additional arms after contacting the item inside of the dome. This behavior is consistent with documented octopus foraging strategies in nature and reaffirms that vision is not essential for successful food foraging and coordinated arm use.

Most studies investigating octopus arm movement have focused on single-arm behavior (Byrne et al., 2006a; Gutnick et al., 2011, 2020; Flash and Zullo, 2023), and observations of coordination

between multiple arms have been limited to studies conducted during visual prey capture and item manipulation (Byrne et al., 2006b; Bidel et al., 2022). When using vision, octopuses preferentially recruit adjacent arms to assist with prey capture, while the recruitment of non-adjacent arms is rare. Octopuses in our study were blindly searching for prey, and adjacent arms (A+1) were recruited most often (*n*=44%). Octopuses also often recruited more distant arms (collectively *n*=56%). The recruitment of non-adjacent arms was most often observed when the recruited arm was physically close to an opening of the dome. It is plausible that, rather than efficiently directed by the neural pathways that interconnect neighboring arms (Graziadei, 1971; Young, 1971; Grasso, 2014), or those that connect non-adjacent arms (Kuuspalu et al., 2022), octopuses occasionally utilize arms based on spatial convenience, suggesting a higher degree of flexibility in their foraging strategies than previously understood. Greater flexibility of arm use may make prey capture more efficient in visually occluded situations.

### Simultaneous use of multiple arms
Octopuses frequently used three, four, or five arms simultaneously to both explore the interior of the dome and manipulate the object inside, while the remaining arms were engaged in maintaining stability and grip on the exterior of the dome. Limiting the simultaneous use of arms to three, four, or five may effectively allow for coordination of explorative and manipulative tasks, conservation of energy, and the ability to reserve other limbs for non-foraging purposes. At the same time, utilizing multiple limbs may also help reduce the likelihood of prey escape tactics, suggesting the functional importance of limb redundancy.

### CNS versus PNS arm control
In nature, octopuses are often observed extending arms into areas likely to contain prey. Similarly, in this experiment, octopuses were observed extending an arm into one of the openings of the dome and using it to explore the interior with a general sweeping motion. In our experiments, the sweeping motions used by octopuses to explore the dome were only moderately effective in establishing contact with the item. It is possible that the seemingly stereotyped searching movements seen during exploration are under control by the PNS. Several researchers have suggested that these types of searching movements, often used by octopuses during exploration and hunting, may require little or no central control and could be performed by the PNS (Sumbre et al., 2001), relying on tactile and chemical information (e.g. Wells, 1978; Gutfreund et al., 1996; Mather, 1998; Buresch et al., 2022).

During the initial approach of each trial, octopuses preferentially inserted one of their four anterior arms into the dome. This suggests that octopuses may have relied on visual input to guide their initial targeting of the dome, consistent with strategies used to locate potential food sources in nature, before shifting to proprioceptive and tactile exploration as the trial progressed in the visually occluded environment. Although octopus proprioception is generally considered limited, some researchers propose that octopuses are capable of proprioceptive learning, citing evidence of central nervous system (CNS) involvement in goal-directed behavior (Gutnick et al., 2020).

The octopuses in our experiments also often used their arms together in complex coordinated movements to manipulate the objects inside of the dome. This type of blind arm coordination is most likely regulated by communication between the CNS and PNS, as proprioception allows for integrated guidance and feedback

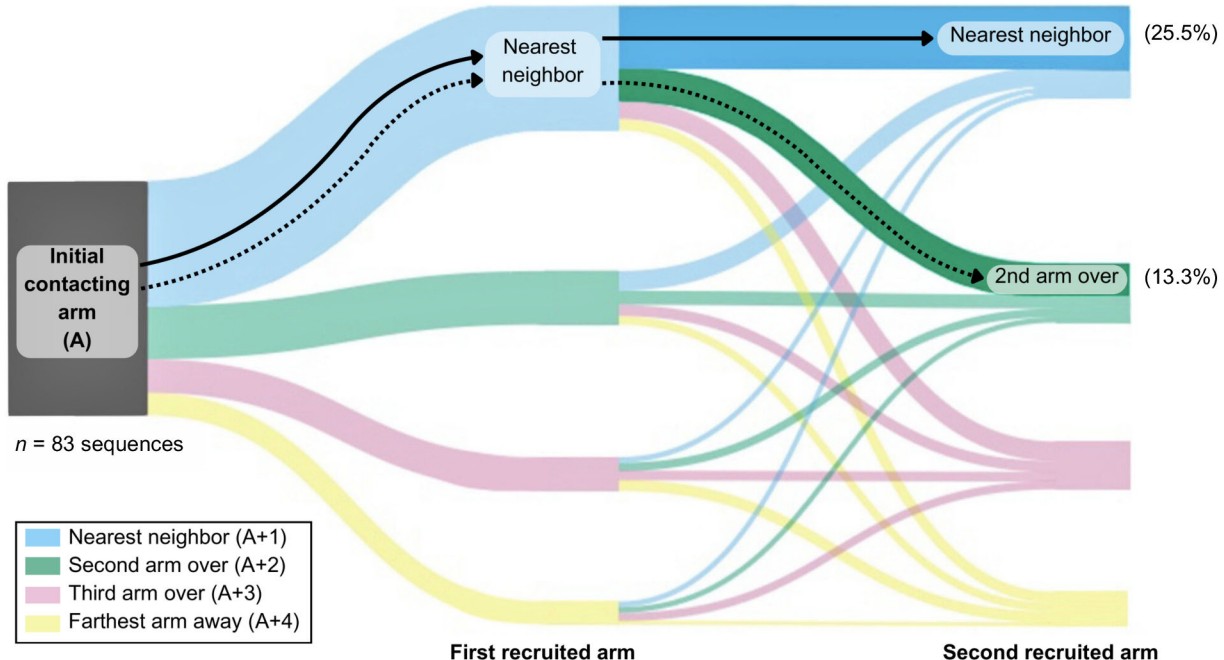

**Fig. 4. Octopuses recruited arms most frequently in two distinct patterns.** When making at least three sequential contacts with the item inside of the dome, octopuses ($n$=14 individual animals; biological replicates) recruited arms in two frequently repeated patterns. The most common transitions were (1) initial contacting arm→nearest neighbor→nearest neighbor (light-blue to dark-blue flow bar; $n$=22, 25.5%), and (2) initial contacting arm→nearest neighbor→second arm over (light-blue to dark-green flow bar; $n$=11, 13.3%). These patterns occurred significantly more frequently than expected by chance (Chi-square goodness-of-fit test, exact two-tailed: $\chi^2$=91.71, $d.f.$=15, $P$<0.001).

between arms (Proske and Gandevia, 2012). Other types of complex arm movements and behaviors have been elicited by direct stimulation of the CNS in partial arm preparations (Zullo et al., 2009), further supporting the involvement of central control in multi-arm behavior. While recent work has illuminated the organization of the octopus nervous system (Chang and Hale, 2023; Olson and Ragsdale, 2023; Neacsu and Crook, 2024; Olson et al., 2025), the extent to which neural pathways may differ across individual arms, such as between anterior and posterior pairs, remains unknown. Our results suggest some positional asymmetry during the initial stages of blind foraging, but do not support

persistent differences in arm use that would indicate functional neural specialization across arm groups.

### Behavioral ecology implications

The octopus's ability to forage successfully and recruit and coordinate multiple arms without reliance on vision offers several advantages. Foraging without complete dependence on sight allows octopuses to search for prey with their arms while visually scanning for approaching predators, enabling them to concurrently manage both offense and defense. Moreover, the ability to utilize all arms equally adds a layer of behavioral flexibility that is particularly advantageous in complex or variable habitats. In our experiments, octopuses often used sweeping arm motions to explore the environment before locating prey-like items, frequently requiring multiple exploratory attempts before establishing initial contact. Even after extracting an item from the dome, they often continued to explore the interior, suggesting persistent spatial searching and possibly an expectation of additional prey. Rather than relying on a fixed motor strategy or designated limbs, octopuses can dynamically adapt their arm usage based on the spatial orientation of the prey, the environment, or the immediate context of the interaction. This high degree of behavioral flexibility in arm use during foraging and item manipulation further enables octopuses to exploit a variety of environments, granting access to prey resources that are inaccessible to visually dependent competitors. Ultimately, this decentralized and adaptable approach to arm use enables octopuses to be highly versatile and efficient predators.

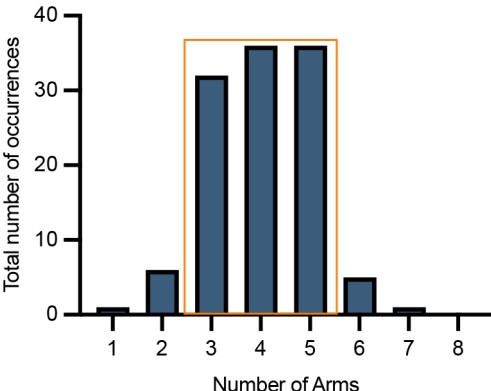

**Fig. 5. Maximum number of arms used simultaneously inside of the dome.** In all trials ($n$=117), octopuses ($n$=14 individual animals; biological replicates) most frequently used three (27.35%), four (30.77%), or five (30.77%) arms simultaneously to explore the dome and/or contact the item inside (orange box; Chi-square goodness-of-fit test: $\chi^2$=134.6, $d.f.$=7, $P$<0.001). No significant differences were found among the use of three, four, or five arms [Wilcoxon signed-rank test (BKY correction): all $q$=1.0, $P$>0.566]. All statistical tests were two-tailed.

### Inspiration for soft robotics

Redundancy in the octopus's body plan could have implications for the design of soft robotic systems based on natural octopus arm behaviors. Octopuses exhibit remarkable dexterity and manipulation capabilities, which have inspired the development of various soft

robotic arms (e.g. Xie et al., 2023; Shih et al., 2023) and fully articulated systems (e.g. Cianchetti et al., 2015). These innovations have the potential for diverse applications, ranging from soft prosthetics and medical technologies to military reconnaissance and search operations. Notably, in our experiment, octopuses showed no clear preference for utilizing individual arms, instead using their arms in a flexible, redundant manner. This lack of arm specialization could be an exciting avenue for bio-inspired robotic design. As soft robotic solutions based on octopus arms continue to evolve, an effective new approach might involve designing systems in which all arms behave similarly, rather than separating the load between axes or insisting on specialization. This redundancy would allow the system to remain functional in the event of a single arm failure, much like how octopuses adapt to varying tasks with no reliance on any single arm.

## MATERIALS AND METHODS
### Subjects
Fourteen *Octopus bimaculoides* were reared at the Marine Biological Laboratory. Eight of the animals were collected off the coast of southern California (74-193 g, age and sex unknown), and six were reared in the laboratory (190-332 g, age range 12-19 months, sex unknown) (Table S9 for details). Individuals were housed separately in 80-l aquaria enriched with clay-pot dens, artificial plants, and children's toys. Aquaria were supplied with a consistent flow of filtered ambient seawater and bubble aeration, and were regulated to 15±0.2°C, pH 8.0±0.1, and a 12/12-h light/dark cycle. Octopuses were fed once daily with live food items including Asian shore crabs (*Hemigrapsus sanguineus*), blue mussels (*Mytilus edulis*) and *Crepidula* spp. snails.

### Artificial rock dome
To mimic a natural environment in which an octopus might forage blindly, an artificial half-dome with four elbow-shaped openings was 3D printed using standard grey resin on a Formlabs Form 3B+ resin printer (Fig. 6). Small rocks were epoxied (MarineWeld™) to the exterior of the dome to imitate a natural crevice and obscure the openings; the elbow-shape further prevented the octopuses from seeing inside of the dome. One open side of the half-dome was secured to a piece of glass to allow observers to film arm behavior inside of the dome. The adjacent open side was attached to a brick, using Velcro®, preventing the octopuses from shifting the dome's position within the experimental tank. The dome was printed in two sizes to account

for size differences between the octopuses. Openings to the dome were only large enough for one to two octopuses' arms to fit through.

### Dome training protocol
Octopuses were trained to feed from the dome in a four-phase process adapted from Buresch et al. (2022) (Fig. 6A-C). Training occurred twice daily, each phase requiring two consecutive days of successful feeding from the dome before advancing. In the event of two consecutive failures the octopus reverted to the previous training phase. In all phases, frozen crab or shrimp were used as the reward item inside of the dome.

In phase one, a dome without a glass barrier was submerged in the octopus's home tank at a 45° angle to the side of the tank. During this phase, the octopus was given an unlimited amount of time to explore the dome and retrieve the reward item. Occasionally, an artificial crab was used to visually lure the animal to the dome. In phase two, the dome without a glass barrier was placed flush to the side of the octopus's tank, and the animal was given a maximum of 2 h to retrieve the reward. In phase three and four, a glass barrier was added to the dome, and the dome was attached to a brick with Velcro®. The time for phase three was limited to 1 h, and the time limit for phase four was reduced to 20 min.

After successfully completing phase four of the training protocol, each octopus advanced to an acclimation phase in which it was moved from the home tank to the experimental arena twice daily and completed the dome feeding task within 20 min. No filming occurred during the acclimation phase. Experimental trials began after the octopus completed a week of successful feedings in the experimental arena. The amount of time for each octopus to progress through the training protocol and acclimation phase was variable.

### Experimental protocol
To evaluate arm use and recruitment across a range of sensory stimuli, five items were tested inside of the dome each differing in tactile, chemosensory, and physical properties relevant to prey recognition: (1) a live Asian shore crab, (2) a frozen Asian shore crab, (3) a 3D-printed crab shape, (4) a 3D-printed rock shape, and (5) a crab-infused agarose disc. Fourteen octopuses were presented with the items in a random order, and each item was presented to ten subjects (Table S9).

The live crab and frozen crab were edible prey items that could be removed from the dome. The 3D-printed crab was a rigid, prey-shaped model that mimicked the mechano-tactile properties of a real crab but lacked chemosensory cues; it was similarly small enough to be removed. The crab agarose item was constructed from edible agarose infused with crab extract, providing chemosensory cues but lacking realistic tactile structure; it was too

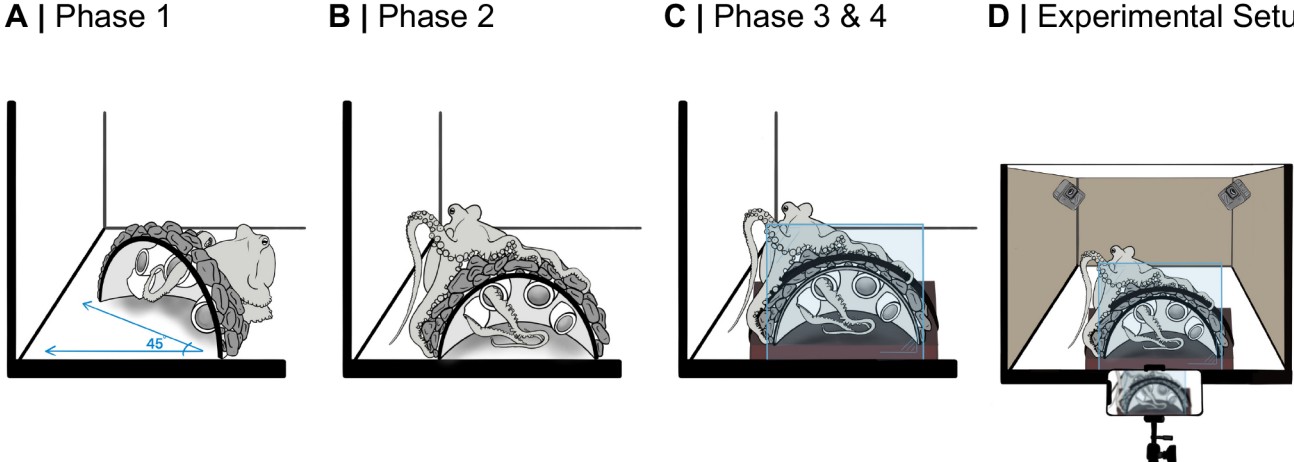

## A | Phase 1   B | Phase 2   C | Phase 3 & 4   D | Experimental Setup

**Fig. 6. 3D-printed half-dome four-phase training protocol and experimental tank setup.** (A) Phase 1: the half-dome was placed at a 45° angle from the side of the octopus's home tank; unlimited time to retrieve reward. (B) Phase 2: the dome was placed flush against the side of the home tank; 2-h time to retrieve reward. (C) Phase 3: a glass barrier was added to the open end of the dome; the dome was attached to a brick with Velcro® and placed flush against the side of the home tank; 1-h time limit to retrieve reward. Phase 4: time to retrieve reward reduced to 20 min. Following phase 4, the octopus was acclimated to an experimental tank and performed the dome task twice daily with a 20-min time limit. (D) Experimental setup. The dome with the glass barrier, attached to a brick, was submerged in an experimental tank. An iPhone 14 mounted on a tripod recorded the dome's interior, while two GoPro HERO10 cameras recorded the left and right external views. Translucent sand colored material covered the tank to minimize human interaction.

Biology Open

large to be removed, allowing extended access during trials. The 3D-printed rock was a non-prey control item that neither felt nor tasted like prey and was also non-removable. Importantly, once an item was removed from the dome, no further contact observations were recorded for that trial.

A 75-l glass aquarium filled with filtered ambient seawater was used as the experimental arena (Fig. 6D). Three sides of the tank were covered with opaque material to minimize disturbance. The tank was drained and refilled prior to each experimental trial to reduce odors from conspecifics or prey items used in the previous trial. For each trial, the artificial dome, containing one of the items selected at random, was submerged in the experimental tank, with the glass side of the dome flush to the transparent side of the tank.

Three video cameras were used to record octopuses' behavior. An iPhone 14 (4K, 60 FPS) mounted to a flexible tripod was used to record the front view of the dome's interior. External left and right views of the dome were captured using two GoPro HERO10 cameras (4K, 30 FPS). The GoPro cameras were submerged at the limit of the air-water interface and mounted to either back corner of the tank using magnets, allowing for maximum angle capture and reduction of light reflection.

Two experimenters set up the experimental arena. One experimenter isolated a back corner of the tank with an opaque barrier while the other transferred the octopus to the isolated corner. Both experimenters started the three cameras synchronously, and the opaque barrier was then removed. A translucent sheet covered the entire experimental arena to minimize human influence, and the octopus was left to explore the dome for 20 min. Experimental trials were conducted 2 days per week, twice per day (morning/afternoon). Individual octopuses completed between four and 12 experimental trials (Table S9).

### Video analysis

Videos from each trial (*n*=117) were scored manually by two observers, and scores were compared for accuracy. If scoring discrepancies were found, observers reviewed footage in tandem or consulted a third observer, if necessary. For each trial, videos were scored for (1) the first arm to enter the inside of the dome in the first approach, (2) the sequence of individual arms used for exploring inside of the dome or contacting the item each time an octopus approached the dome, (3) the recruitment type between any two arms when sequentially contacting the item inside of the dome, (4) the sequence of recruitment types between the first three arms used to contact the item, and (5) the maximum number of arms used simultaneously for exploring inside of the dome and/or contacting the item in each trial. Arm

use and recruitment were graded for every approach in each trial, which was defined as the interval beginning with the insertion of the first arm into the dome and ending with the animal disengaging from the dome's exterior.

### Arm use and recruitment grading scheme

Arms were labeled according to their anatomical position on the octopus (R1 to L4) and classified as either engaging in exploration or as contacting the item. An arm that was inserted into the dome without contacting the item for at least 3 s was considered an exploration, while an arm that touched or manipulated the item inside of the dome was considered a contact. The arm label and classification defined arm use.

Fig. 7 illustrates scoring for arm recruitment. Arm recruitment was defined as the engagement of an additional arm after an initial arm contacted the item inside of the dome. To determine the arm recruitment type, arms were labeled according to the initial arm to contact the item, which was always labeled 'A'. The next arm to contact was scored according to its anatomical position from 'A' (i.e. the number of arms away from the initial arm). This score defined the recruitment type and was used for both clockwise and counterclockwise directions. Following this score, the recruited arm would be reassigned as arm 'A', and this pattern of scoring would repeat for every subsequent arm used to contact the item. Four types of arm recruitments were recorded: A+1, A+2, A+3, and A+4.

Arm recruitment patterns refer to the multiple sequential arm recruitments, focusing on sequences involving three or more arms used to contact the item inside of the dome. To assess patterns of arm recruitment, the sequence of recruitment types between the first three arms used to contact the item in each approach were analyzed. Two recruitments are made when utilizing three arms in sequence (first contacting arm → first recruited arm → second recruited arm). There are 16 possible sequences of arm recruitment between three contacting arms. Sequences that included repeated use of the same arm were excluded, as a new arm was not recruited (e.g. R1 → R2 → R2).

### Data analyses

Data were visualized and statistically analyzed using GraphPad Prism 10 (GraphPad Prism version 10.4.1 for Mac; GraphPad Software, Boston, MA, USA; www.graphpad.com). To assess the effects of octopus origin (wild caught versus laboratory reared), and item presented on arm use and recruitment type, data were analyzed using negative binomial generalized linear mixed models, with octopus identity included as a random effect. As these models revealed no significant effects of individual identity, origin, or

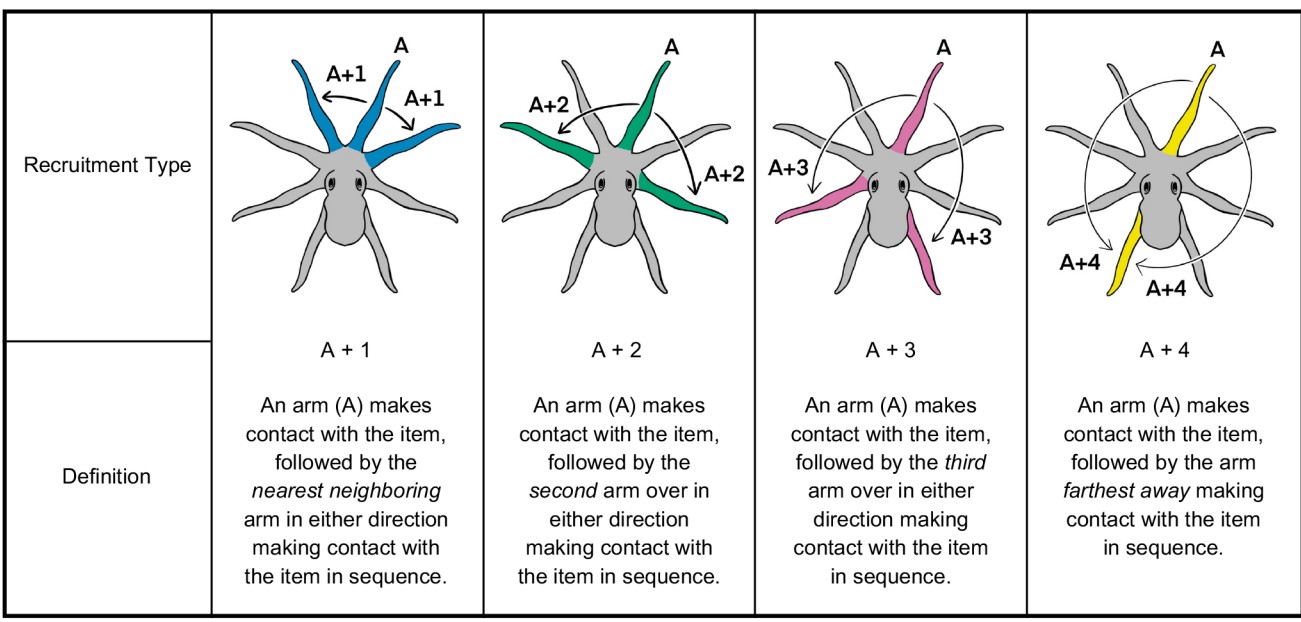

**Fig. 7. Definitions of arm recruitment types.** Diagrams illustrate the four types of observed arm recruitments. The example here is based on arm R1 as the initial arm used, 'A', to contact the item. However, any initial arm used to contact the item was assigned as 'A', and the next arm used to contact the item defined the recruitment type.

item, data were collapsed across trials and analyzed using Friedman's ANOVA. When Freidman's ANOVAs revealed significant differences, pairwise comparisons were performed using Wilcoxon signed-rank tests with false discovery rate correction (Benjamini, Krieger and Yekutieli method).

In additional analyses, frequency distributions of recruitment sequence patterns and simultaneous arm use were evaluated using Chi-square goodness-of-fit tests. When comparing arm use across arm groups (e.g. anterior versus posterior), Wilcoxon signed-rank tests were used to assess paired differences in behavior within individual animals.

## Ethical statement
Cephalopods are not included in laboratory animal welfare regulations in the United States; however, the protocol in this study was conducted in accordance with the Marine Biological Laboratory's Cephalopod Care Policy.

## Acknowledgements
We thank all members of the Hanlon laboratory for their help with animal husbandry and data collection, with special thanks to Jamie Suesser, Trisha Anand, and Emma Campbell. We thank Jean Boal for assistance with statistical analyses. We also thank all members of the Marine Resources Department at the Marine Biological Laboratory for their assistance with water quality measurements and seawater system maintenance.

## Competing interests
The authors declare no competing or financial interests.

## Author contributions
Conceptualization: C.B., K.C.B., A.S.R., R.T.H.; Data curation: A.J.G., C.B., K.C.B., A.S.R., H.M.; Formal analysis: A.J.G., C.B., K.C.B., H.M.; Funding acquisition: K.C.B., R.T.H.; Investigation: A.J.G., C.B., A.S.R., H.M.; Methodology: C.B., K.C.B., A.S.R.; Project administration: A.J.G., C.B., K.C.B.; Resources: K.C.B., R.T.H.; Supervision: A.J.G., K.C.B., R.T.H.; Validation: K.C.B., R.T.H.; Visualization: A.J.G., C.B.; Writing – original draft: A.J.G., K.C.B.; Writing – review & editing: A.J.G., K.C.B., R.T.H.

## Funding
This research was funded by the Office of Naval Research (N00014-21-S-B001). We appreciate partial funding from the Sholley Family Foundation and the Ben-Veniste Family Foundation. Open Access funding provided by Marine Biological Laboratory. Deposited in PMC for immediate release.

## Data and resource availability
All relevant data can be found within the article and its supplementary information.

## First Person
This article has an associated First Person interview with the first author of the paper.

## Peer review history
The peer review history is available online at https://journals.biologists.com/bio/lookup/doi/10.1242/bio.062011.reviewer-comments.pdf.

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
