## [Peer Review File · Biology Open]

How octopuses use and recruit additional arms to find and manipulate visually hidden items

Camille Boucaud, Kendra C. Buresch, Allison Sayre Rooney, Halia Morris, Roger T. Hanlon and Ashley J. Gendreau

DOI: 10.1242/bio.062011

Editor: Lewis Halsey

Review timeline

Original submission:	1 April 2025
Editorial decision:	5 April 2025
First revision received:	13 June 2025
Accepted:	16 June 2025

Original submission

First decision letter

MS ID#: bio.062011

MS TITLE: How octopuses use and recruit additional arms to find and manipulate visually hidden items

AUTHORS: Camille Boucaud; Kendra C. Buresch; Allison Sayre Rooney; Halia Morris; Roger T. Hanlon; Ashley J. Gendreau

I have now reached a decision on the above manuscript.

The reviewer reports are shown at the bottom of this email or can be accessed, together with a copy of this decision letter, by going to:

As you will see, the reviewers raised a number of substantial criticisms that prevent me from accepting the paper at this stage.

They suggest, however, that a revised version might prove acceptable, if you can address their concerns. If you think that you can deal satisfactorily with the criticisms on revision, I would be pleased to see a revised manuscript. We would then return it to the reviewers.

At this stage, we also ask you to ensure your manuscript complies with our formatting guidelines. Provided you are able to fully address the referees' comments, we are positive about publication of your paper (we accept over 95% of revision submissions) and therefore hope you won't mind any extra work involved in reformatting your manuscript at this point.

Please ensure that you clearly highlight all changes made in the revised manuscript. Please avoid using 'Tracked changes' in Word files as these are lost in PDF conversion.

I should be grateful if you would also provide a point-by-point response detailing how you have dealt with the points raised by the reviewers in the 'Response to Reviewers' box. Please attend to all of the reviewers' comments. If you do not agree with any of their criticisms or suggestions please explain clearly why this is so.

Reviewer 1

Comments to Author

In this study, the authors quantified patterns of arm use and recruitment when foraging for food in a visually-occluded environment in octopuses under laboratory conditions. They found that octopuses use all eight arms but tend to preferentially recruit their nearest-neighboring arm when searching for or interacting with food, indicating high flexibility in how they use arms. Thus, in addition to vision, this experiment shows that flexibility in arm use might enable octopuses to successfully forage also in situations where they cannot rely on vision. Overall, the results are interesting, but there are several points in experimental setting and data analyses that were not sufficiently explained or justified.

INTRODUCTION

Lines 70-72: in this paragraph that just comes before the aims of the study, we are said that mechanisms by which octopuses coordinate multiple arms or deploy individual arms for visually-occluded foraging tasks remain largely unknown. The study does not, however, address any endogenous mechanisms through which arms are coordinated, so I think this statement is misplaced. I would rather say something here about why it is important to characterize patterns of arm use and recruitment.

MATERIAL AND METHODS

The study involved both lab-reared octopuses and wild-caught octopuses. I wonder if this influences the response to the tasks. They have probably accumulated over time different experiences, so that they might rely on different patterns of arm use and recruitment. This point needs to be clarified.

Training protocol: this is something novel to some extent for the octopuses, so that their patterns of arm use and recruitment might not be the same as those observed when dealing with familiar environments. I think that it would be interesting to provide some more information on this, possibly with some statistical models to show how they behave.

Line 363: the experiment involved different items to simulate different sensory stimuli. However, I cannot see any test about how patterns of arm use and recruitment are influenced by the item. I think that this is important to address given the large among item differences (e.g. live vs. frozen prey).

Lines 391-397: I wonder if octopuses use arms the same way, e.g. do all enter the same extent inside the dome? Or some just touch the entrance for exploration?

Data analyses: I am unsure about the way data have been handled for the statistics. You have individuals tested multiple times, and each octopus used several arms. In both case, data are not independent from each other, and this may raise a problem of pseudoreplication. Wouldn't it be better to rely on linear mixed models where you can control for individual identity? I also wonder if individual patterns of arm use and recruitment were consistent across trials.

RESULTS

Did the angle from which octopuses approached the dome influence their patterns of arm use and recruitment?

Line 101: in line with what I said about statistical models, I wonder if the arm used in subsequent approaches was dependent on which arm was used in the first approach, e.g. is it more likely for an octopus that uses the right anterior arm in the first approach to use the left anterior arm in the second approach or the posterior right arm?

Reviewer 2

Comments to Author

General comments:

This is a nice study that examines octopus' choice of first arm use and subsequent arms in searching areas that they can't see. The experimental procedure is well thought out and the approach is clear. The figures are wonderful, well done. I mostly have minor suggestions for change or clarification that I believe will help the reader understand the work more easily. I think some of the discussion text could be moved to the introduction as background information, which may leave the discussion a little lacklustre (in which case I suggest the authors think broadly about alternate explanations, experiments or scenarios where their results may or may not apply, including comparison to other octopus species or any other niches that octopus species might occupy in a different habitat). The two most major and direct comments are listed below.

It's not immediately clear if the results refer to all octopus as a cohort, or individually. I.e. when all data were pooled for the 15 octopuses, no individual arm was preferred, but did any of the individual octopuses have their own unique arm preference? Similar to left or right-handed in humans, do individual octopus have preferred arms? I suspect the authors know the answer already without doing more analysis but I think it should be made clearer in the paper, or explain why it's not important/interesting/relevant.

I also think the manuscript would benefit from clarity on the definition of 'first approach' versus subsequent approaches. It wasn't 100% clear to me if the subsequent approach refers to the next food item placed in the dome, or a second attempt at finding the item after a failed first attempt.

Overall a good study, just please tighten up a few aspects in the writing as per my comments.

Specific comments:

Ln 62: Minor point but it's slightly ambiguous if this means 330 million or 330 exactly.

Ln 70: This "gap" statement sounds like it sets up the study to determine the "mechanisms by which octopus coordinate multiple arms" which I don't think this is what it does and doesn't flow exactly to the next paragraph about exploring patterns, relationships etc - but not necessarily neural mechanisms. Just a minor change in wording at Ln 70 would help set up the study more clearly.

Ln 75: what is the difference between arm use and recruitment?

Ln 78-79: what is "arm recruitment type"?

Ln 79: What are "recruitment patterns"? I also think "among" is the wrong word here, perhaps "used in"?

Ln 96-103: It's not immediately clear if this refers to all octopus as a cohort, or individually. I.e. when all data were pooled for the 15 octopuses, no individual arm was preferred, but did any of the individual octopuses have their own unique arm preference?

Ln 155: "expected" by what? Prior hypotheses or more frequently than expected if it was random?

Ln 160-171: Were there any relationships between number of arms recruited and time taken to retrieve the item? I.e. did the octopus start with 3-5 arms at once, or were arms added sequentially as the search failed? E.g. I'm picturing the animal that used 7 arms at once as being 'frustrated' that it couldn't retrieve the item!

Ln 174-186: This whole paragraph is better suited to the introduction and would provide valuable background information that I found somewhat lacking in the introduction.

Ln 202-210: I think this information could be moved to the introduction as relevant background for the rationale of the study.

Ln 289-296: I'm not sure this small section adds much value to the manuscript. How does it discuss the results of the study? It could go into the introduction because it doesn't discuss any new information discovered by the present study. I suggest either a) moving it to the introduction, b) significantly expanding it with reference to the results, or c) deleting it.

Ln 298-310: This section feels a little contrived. It's probably fine to leave it, but it should be the first section to be cut if anything needs to be removed to make room for more pressing information as requested by reviewed.

Ln 344: Either "In phase one" or "In the first phase"

Ln 385-386: Is the "behavioural assay" the exact same as the "experimental protocol". Best to be consistent with terminology. Also it doesn't say for how many days this was followed - i.e. how many trials for each animal?

Ln 423: I believe Kruskal-Wallis requires independent samples, but multiple observations from the same individuals were pooled together - please check this point.

Ln 592 (Fig 4): This image quality is noticeably worse than the others (but may be a side effect of pdf rendering).

Reviewer's Responses to Questions

Experimental quality

Does each figure have the proper controls?

If 'No', please indicate reasons in Comments for Author box below.

Reviewer #1:

- No

Reviewer #2:

- Yes

Were the data analyzed using appropriate statistical tests?

If 'No', please indicate reasons in Comments for Author box below.

Reviewer #1:

- No

Reviewer #2:

- Yes

Reproducibility

Were experiments performed using adequate number of biological replicates?

If 'No', please indicate reasons in Comments for Author box below.

Reviewer #1:

- Yes

Reviewer #2:

- Yes

Does the methods section provide sufficient detail to permit reproducibility?

If 'No', please indicate reasons in Comments for Author box below.

Reviewer #1:

- Yes

Reviewer #2:

- Yes

Completeness

Are the manuscript's conclusions supported by the data?

If 'No', please indicate reasons in Comments for Author box below.

Reviewer #1:

- No

Reviewer #2:

- Yes

Scholarship

Do the authors cite and discuss the merits of data that would argue for and against their conclusion?

If 'No', please indicate reasons in Comments for Author box below.

Reviewer #1:

- Yes

Reviewer #2:

- Yes

Does the manuscript title & abstract accurately reflect the contents of the manuscript, without hyperbole?

If 'No', please indicate reasons in Comments for Author box below.

Reviewer #1:

- Yes

Reviewer #2:

- Yes

First revision

Point-by-point response to comments

Reviewer Comments:

Reviewer 1: In this study, the authors quantified patterns of arm use and recruitment when foraging for food in a visually-occluded environment in octopuses under laboratory conditions. They found that octopuses use all eight arms but tend to preferentially recruit their nearest-neighbor arm when searching for or interacting with food, indicating high flexibility in how they use arms. Thus, in addition to vision, this experiment shows that flexibility in arm use might enable octopuses to successfully forage also in situations where they cannot rely on vision. Overall, the results are interesting, but there are several points in experimental setting and data analyses that were not sufficiently explained or justified.

INTRODUCTION

Lines 70-72: in this paragraph that just comes before the aims of the study, we are said that mechanisms by which octopuses coordinate multiple arms or deploy individual arms for visually-occluded foraging tasks remain largely unknown. The study does not, however, address any endogenous mechanisms through which arms are coordinated, so I think this statement is misplaced. I would rather say something here about why it is important to characterize patterns of arm use and recruitment.

We appreciate this observation. We have revised the phrasing to avoid implying that this study addresses endogenous mechanisms of arm control. Instead, we now emphasize that there is limited characterization in the literature of how octopuses coordinate multiple arms or deploy individual arms during visually occluded foraging. We feel that this more accurately aligns with the scope and contributions of our work.

MATERIAL AND METHODS

The study involved both lab-reared octopuses and wild-caught octopuses. I wonder if this influences the response to the tasks. They have probably accumulated over time different experiences, so that they might rely on different patterns of arm use and recruitment. This point needs to be clarified.

This is an interesting question, and we followed it up by using a negative binomial generalized linear mixed model to test whether octopus origin influenced patterns of arm use or recruitment. Our analysis revealed no significant effects of origin on either behavior. We have now clarified this result in the Methods sections of the manuscript.

Training protocol: this is something novel to some extent for the octopuses, so that their patterns of arm use and recruitment might not be the same as those observed when dealing with familiar environments. I think that it would be interesting to provide some more information on this, possibly with some statistical models to show how they behave.

This is an insightful suggestion; however, this experiment was intentionally designed to approximate a naturalistic foraging scenario. Tactile feeding strategies are central to octopuses, and in nature octopuses are often observed extending their arms blindly into crevices, engaging in what has been described as a “speculative groping” approach, to locate prey within coral heads or

along rocky substrates. The artificial rock dome used in this study was designed to reflect these naturalistic conditions. While a direct comparison between arm use in familiar versus novel environments would indeed be valuable, we believe such an analysis falls outside the scope of this current paper.

Line 363: the experiment involved different items to simulate different sensory stimuli. However, I cannot see any test about how patterns of arm use and recruitment are influenced by the item. I think that this is important to address given the large among item differences (e.g. live vs. frozen prey).

To address this concern, we conducted a negative binomial generalized linear mixed model with the item as a fixed effect. These analyses revealed no significant effect of item type on either arm use or recruitment patterns. We have now explicitly included this result in the Methods sections of the revised manuscript.

Lines 391-397: I wonder if octopuses use arms the same way, e.g. do all enter the same extent inside the dome? Or some just touch the entrance for exploration?

In our analysis, arms were classified either as engaging in exploration (entering the dome without contacting the item for more than 3 seconds) or as making item contact (with suckers in direct contact with the item). Only arm entries into the interior of the dome were included in these classifications. Touches to the exterior of the dome or rim of the entrances were not considered exploratory behaviors and were therefore excluded from analysis. The video analysis section has been reworded to clarify how these behaviors were analyzed.

Data analyses: I am unsure about the way data have been handled for the statistics. You have individuals tested multiple times, and each octopus used several arms. In both case, data are not independent from each other, and this may raise a problem of pseudoreplication. Wouldn't it be better to rely on linear mixed models where you can control for individual identity? I also wonder if individual patterns of arm use and recruitment were consistent across trials.

We appreciate the concern regarding the potential for pseudo-replication and the importance of accounting for repeated measures. To address this, we used negative binomial generalized linear mixed models that included individual octopus identity as a random effect. These models were used to test whether arm use and recruitment patterns varied as a function of individual identity, item type, or origin (wild vs. lab-reared). Our analysis found no significant effects of any of these variables. Based on these findings, and to assess overall behavioral patterns at the group level, we then collapsed the data by individual and used non-parametric, repeated-measures tests (e.g., Friedman's test and Wilcoxon signed-rank tests) for group-wise comparisons. These steps have been clarified in the Methods section of the revised manuscript.

RESULTS

Did the angle from which octopuses approached the dome influence their patterns of arm use and recruitment?

This is an intriguing thought. While the angle of approach was noted informally during observations, it appeared to vary across trials and individuals in a seemingly random manner. As such, it was not systematically recorded or included in the analysis. We agree that future studies incorporating the approach angle as a factor could provide additional insights into arm use and recruitment dynamics.

Line 101: in line with what I said about statistical models, I wonder if the arm used in subsequent approaches was dependent on which arm was used in the first approach, e.g. is it more likely for an octopus that uses the right anterior arm in the first approach to use the left anterior arm in the second approach or the posterior right arm?

This is a thoughtful question. In our study, an approach was defined as the interval beginning with the insertion of the first arm into the dome and ending with the animal disengaging from the dome's exterior. Each approach was treated as an independent event, as octopuses typically moved away - by crawling or swimming - before initiating a subsequent approach. As such, the identity of

the first arm used in one approach was not considered predictive of the first arm used in subsequent approaches within the same trial. We have edited the video analysis section to clarify the definition of an approach.

Reviewer 2: General comments:

This is a nice study that examines octopus' choice of first arm use and subsequent arms in searching areas that they can't see. The experimental procedure is well thought out and the approach is clear. The figures are wonderful, well done. I mostly have minor suggestions for change or clarification that I believe will help the reader understand the work more easily. I think some of the discussion text could be moved to the introduction as background information, which may leave the discussion a little lacklustre (in which case I suggest the authors think broadly about alternate explanations, experiments or scenarios where their results may or may not apply, including comparison to other octopus species or any other niches that octopus species might occupy in a different habitat). The two most major and direct comments are listed below.

It's not immediately clear if the results refer to all octopus as a cohort, or individually. I.e. when all data were pooled for the 15 octopuses, no individual arm was preferred, but did any of the individual octopuses have their own unique arm preference? Similar to left or right-handed in humans, do individual octopus have preferred arms? I suspect the authors know the answer already without doing more analysis but I think it should be made clearer in the paper, or explain why it's not important/interesting/relevant.

This is an important clarification. As noted in our response to Reviewer 1, we reanalyzed our data using negative binomial GLMMs to test whether individual octopus identity predicted differences in arm use or recruitment patterns. These models did not reveal significant individual-level effects. Based on this, and to explore population-level behavioral patterns, we summarized arm use across octopuses using repeated-measures non-parametric tests. We have clarified in the revised manuscript that our analyses first accounted for individual variability, and that no evidence of individual arm preference (e.g., consistent lateralization) was found.

I also think the manuscript would benefit from clarity on the definition of 'first approach' versus subsequent approaches. It wasn't 100% clear to me if the subsequent approach refers to the next food item placed in the dome, or a second attempt at finding the item after a failed first attempt.

While the definition of "approach" was provided in the methods section under "Video analysis," we agree that further clarification was needed. We have revised the text to more clearly define this term.

Overall a good study, just please tighten up a few aspects in the writing as per my comments.

Specific comments:

Ln 62: Minor point but it's slightly ambiguous if this means 330 million or 330 exactly.

We clarified this number as it is 330 million and not 330 exactly.

Ln 70: This "gap" statement sounds like it sets up the study to determine the "mechanisms by which octopus coordinate multiple arms" which I don't think this is what it does and doesn't flow exactly to the next paragraph about exploring patterns, relationships etc - but not necessarily neural mechanisms. Just a minor change in wording at Ln 70 would help set up the study more clearly.

Reviewer 1 was also concerned with this. As indicated above, we have revised the phrasing to avoid implying that this study addresses endogenous mechanisms of arm control. Instead, we now emphasize that there is limited characterization in the literature about how octopuses coordinate multiple arms or deploy individual arms during visually occluded foraging.

Ln 75: what is the difference between arm use and recruitment?

Ln 78-79: what is "arm recruitment type"?

Ln 79: What are "recruitment patterns"? I also think "among" is the wrong word here, perhaps "used in"?

We understand that some of these terms may require further clarification. The term "arm use" refers to the individual arm used to explore the interior of the dome or make contact with the item. "Arm recruitment" describes the engagement of an additional arm after the first arm has contacted the item. The term "arm recruitment type" categorizes this sequence, identifying how arms are used in relation to each other (e.g. when an octopus contacted the item with R1 then R2 sequentially, it was termed a "nearest-neighbor" or "A+1" recruitment type). Lastly, "recruitment patterns" refers to the coordination of three or more arms during these interactions, outlining how arms are engaged in a sequential or coordinated manner. While the definitions of "arm recruitment" and "recruitment patterns" were detailed in the Methods section under "Arm recruitment grading scheme," we have revised the introduction to provide additional clarity on these definitions and edited the Methods section to further explain the concept of "arm use," "recruitment type," and "recruitment patterns."

Ln 96-103: It's not immediately clear if this refers to all octopus as a cohort, or individually. I.e. when all data were pooled for the 15 octopuses, no individual arm was preferred, but did any of the individual octopuses have their own unique arm preference?

As stated above, we reanalyzed the data using negative binomial GLMMs and found no significant individual-level differences in arm use. The revised manuscript clarifies that individual variability was accounted for and no consistent arm preference was detected.

Ln 155: "expected" by what? Prior hypotheses or more frequently than expected if it was random?

We clarified that this occurred more frequently than expected by chance.

Ln 160-171: Were there any relationships between number of arms recruited and time taken to retrieve the item? I.e. did the octopus start with 3-5 arms at once, or were arms added sequentially as the search failed? E.g. I'm picturing the animal that used 7 arms at once as being 'frustrated' that it couldn't retrieve the item!

This is an interesting question. We did not specifically explore the relationship between the number of arms used simultaneously and the time taken to retrieve the item. Simultaneous arm use refers to the maximum number of arms used at one time inside of the dome that were either engaged in exploration or item contacting inside of the dome during the entirety of each trial. It is important to note that not all the arms were necessarily involved in manipulating the item.

Ln 174-186: This whole paragraph is better suited to the introduction and would provide valuable background information that I found somewhat lacking in the introduction.

We agree that the content of this paragraph provides useful background information. While we believe it serves an important role in framing the discussion section, we understand your concern regarding potential redundancy with the introduction. We have edited the text in both the introduction and discussion to reduce overlap and ensure that each section contributes distinct and complementary context to the manuscript.

Ln 202-210: I think this information could be moved to the introduction as relevant background for the rationale of the study.

As with the previous comment, we have revised the text to incorporate some of this information into the introduction, while retaining key elements in the discussion to help frame that section effectively.

Ln 289-296: I'm not sure this small section adds much value to the manuscript. How does it discuss the results of the study? It could go into the introduction because it doesn't discuss any new information discovered by the present study. I suggest either a) moving it to the introduction, b) significantly expanding it with reference to the results, or c) deleting it.

We agree that the original version of this section did not sufficiently connect to the findings of our study. In response, we have revised the paragraph to more directly reference our results on equal arm use and to expand on its relevance to behavioral flexibility and foraging success.

Ln 298-310: This section feels a little contrived. It's probably fine to leave it, but it should be the first section to be cut if anything needs to be removed to make room for more pressing information as requested by reviewed.

We understand your concern regarding the emphasis on soft robotics, however we believe that this section appeals to the broad readership of *Biology Open* by highlighting the potential applications of octopus-inspired behaviors in bio-robotics, which may engage not only those interested in animal behavior but also those in related fields like robotics, engineering, and medical innovation. By drawing on this connection, we hope to bridge the gap between biological research and interdisciplinary applications. However, we have edited the text to better relate to our findings.

Ln 344: Either "In phase one" or "In the first phase"

Thank you for catching this. We have removed the extra word "the," so it now reads "In phase one."

Ln 385-386: Is the "behavioural assay" the exact same as the "experimental protocol". Best to be consistent with terminology. Also it doesn't say for how many days this was followed - i.e. how many trials for each animal?

"Behavioral assay" and "experimental protocol" refer to the same procedure. We have revised the text for consistency, using the term "experimental trials." Additionally, we have included the number of trials each animal completed, which ranged from four to twelve trials, and added a supplementary table that outlines the details.

Ln 423: I believe Kruskal-Wallis requires independent samples, but multiple observations from the same individuals were pooled together - please check this point.

Thank you for identifying this important statistical point. We agree that the Kruskal-Wallis test assumes independent samples and was not appropriate for the repeated-measures data. We have revised the analysis accordingly and now use Friedman's test, which accounts for repeated measures within individuals. This correction has been implemented throughout the manuscript and figure captions, and full statistical details are provided in the revised Materials and Methods section.

Ln 592 (Fig 4): This image quality is noticeably worse than the others (but may be a side effect of pdf rendering).

We also noticed the image quality issue on Figure 4. This is indeed a side effect of the PDF rendering. The image appears clear in the individual figure file PDF.

Second decision letter

MS ID#: bio.062011R1

MS TITLE: How octopuses use and recruit additional arms to find and manipulate visually hidden items

AUTHORS: Camille Boucaud; Kendra C. Buresch; Allison Sayre Rooney; Halia Morris; Roger T. Hanlon; Ashley J. Gendreau

I received your resubmission at a helpful time and have been able to fully read your rebuttal and associated manuscript edits straight away. I am pleased with your responses and am happy to tell you that your manuscript has been accepted for publication in Biology Open, pending our standard publication integrity checks. It was accepted on 16 Jun 2025.